# BDNF and NGF Expression in Preneoplastic Cervical Disease According to HIV Status

**DOI:** 10.3390/ijms241310729

**Published:** 2023-06-27

**Authors:** Angelo Sirico, Saverio Simonelli, Sara Pignatiello, Caterina Fulgione, Laura Sarno, Francesco Chiuso, Giuseppe Maria Maruotti, Matilde Sansone, Maurizio Guida, Luigi Insabato

**Affiliations:** 1Department of Neuroscience, Reproductive Sciences and Dentistry, University of Naples Federico II, 80131 Naples, Italy; siricoangelo@gmail.com (A.S.); caterina.fulgione@gmail.com (C.F.); gm.mar@tiscali.it (G.M.M.); sansonemati@gmail.com (M.S.); profmaurizioguida@gmail.com (M.G.); 2Pathology Unit, Department of Advanced Biomedical Sciences, University of Naples Federico II, 80131 Naples, Italy; saverio.simonelli.95@gmail.com (S.S.); g.insabato@gmail.com (L.I.); 3Anatomic Pathology Unit, Ospedale del Mare, 80147 Naples, Italy; sarapignatiello91@libero.it; 4Department of Molecular Medicine and Medical Biotechnology, University of Naples Federico II, 80131 Naples, Italy; francesco.chiuso@gmail.com

**Keywords:** neurotrophins, BDNF, NGF, HIV, HPV, cervical cancer, inflammation

## Abstract

Background. Neurotrophins, such as BDNF and NGF, are overexpressed in tumor cells in cervical cancer, and HIV infection is associated with the upregulation of neurotrophin expression. Therefore, we aimed to investigate whether BDNF and NGF are overexpressed in preneoplastic cervical disease from HIV-infected women. Methods. Women with preneoplastic cervical lesions (cervical intraepithelial neoplasia grade 2 or 3) were prospectively enrolled and grouped according to their HIV status. Samples from Loop Electrosurgical Excision Procedure (LEEP) for suspected cervical cancer were obtained, and immunohistochemistry was performed to evaluate BDNF and NGF expression. Results. We included in our analysis 12 HIV-infected patients who were matched with 23 HIV-negative patients as a control group. Immunohistochemistry analysis showed that BDNF expression was significantly higher in cervical preneoplastic lesions from HIV-positive women than in the lesions from the control group. In particular, BDNF was expressed in 8/12 HIV-positive patients and 7/23 HIV-negative patients (66.7% vs. 30.4%, χ^2^ = 4.227; *p* = 0.040). NGF expression was not significantly higher in cervical preneoplastic lesions from HIV-positive women compared with that in the lesions from the control group. In particular, NGF was expressed in 8/12 HIV-positive patients and in 12/23 HIV-negative patients (66.7% vs. 52.2% χ^2^ = 0.676; *p* = 0.411). Logistic regression analysis showed that the HIV status is an independent predictor of BDNF expression in pre-invasive preneoplastic cervical disease when considered alone (crude OR 4.6, 95% CI 0.027–20.347; *p* = 0.046) and when analyzed with other co-factors (adjusted OR 6.786, 95% CI 1.084–42.476; *p* = 0.041). Conclusions. In preneoplastic cervical disease, BDNF expression is higher in HIV-infected women than in non-infected controls, and this is independent of the clinical features of the patients and from the presence of the HPV-HR genotype. BDNF can play a key role as a link between the pathways by which HIV and HPV interact to accelerate cervical cancer progression and invasion. These data can be useful to better understand the role of neurotrophins in the cancerogenesis of cervical cancer and the possible therapeutic strategies to improve disease outcomes.

## 1. Introduction

Cervical cancer remains a significant public health issue, causing approximately 0.6 million cases and 0.3 million deaths annually. It is currently the fourth most prevalent cause of both cancer occurrence and mortality among women globally [1]. Almost all cases of cervical cancer are caused by a persistent infection of the cervix with high-risk genotypes of human papillomavirus (HPV), specifically HPV-16 and HPV-18 [2]. However, it is important to note that the virus itself is not the sole factor responsible for the development of this cancerous condition. The dysregulation of viral and host gene expression, which occurs as a result of viral DNA integration into the host cell’s genome, along with epigenetic modifications, plays a critical role in the process of carcinogenesis [3]. 

The impact of immunodeficiency caused by HIV on female genital HPV is multifaceted and involves various effects such as increased susceptibility to infection, co-infection with multiple HPV genotypes, persistent infection, reactivation of dormant infections, and a higher risk of developing precancerous and cancerous conditions [4]. Women living with HIV have a greater likelihood of developing HPV-related cervical lesions compared with women who do not have HIV. The prevalence of these lesions is estimated to be three times higher in HIV-positive women, particularly due to decreased levels of CD4+ T lymphocytes (a key immune cell) and higher viral loads associated with HIV infection [5]. Although the link between HIV infection and cervical cancer is confirmed from published epidemiological studies, the exact biological mechanisms underlying the role of HIV infection in the initiation and progression of cervical preneoplastic disease are not yet established.

Neurotrophins, a family of growth factors involved in neuronal development and survival, have emerged as significant players in the development and progression of cervical cancer [6,7,8]. Among the neurotrophins, brain-derived neurotrophic factor (BDNF) has received considerable attention for its role in cervical cancer [9]. Studies have demonstrated that BDNF and its receptor TRKB are overexpressed in cervical cancer tissues compared with normal tissues. This overexpression has been associated with advanced stages of the disease, suggesting a potential involvement in tumor progression [10].

BDNF exerts its effects by binding to TRKB receptors, initiating signaling pathways that promote cell survival, proliferation, migration, and invasion. Activation of the ERK and AKT pathways by BDNF/TRKB signaling has been implicated in cervical cancer cell proliferation. The ERK pathway plays a crucial role in cell survival and proliferation by regulating the expression of genes involved in these processes. AKT signaling, on the other hand, promotes cell survival by inhibiting apoptosis. BDNF/TRKB-mediated activation of these pathways contributes to increased cell growth, resistance to apoptosis, and the ability of cancer cells to invade surrounding tissues.

In addition to its role in cell proliferation and survival, BDNF/TRKB signaling has been associated with the epithelial–mesenchymal transition (EMT), a process involved in tumor metastasis. The EMT allows cancer cells to acquire invasive properties, enabling them to detach from the primary tumor and migrate to distant sites. BDNF/TRKB signaling has been shown to induce EMT-related changes in cervical cancer cells, including the downregulation of E-cadherin (a cell adhesion molecule) and the upregulation of mesenchymal markers such as N-cadherin and vimentin. These alterations promote cell motility and invasiveness, contributing to the metastatic potential of cervical cancer.

Furthermore, BDNF has been implicated in angiogenesis, the formation of new blood vessels that provide nutrients and oxygen to growing tumors. BDNF/TRKB signaling has been shown to induce the expression of vascular endothelial growth factor (VEGF), a key mediator of angiogenesis, in cervical cancer cells. This suggests that BDNF may play a role in facilitating the formation of new blood vessels to support tumor growth and metastasis.

HIV infection has been shown to have profound effects on the expression and regulation of neurotrophins. HIV-associated immunodeficiency can lead to alterations in neurotrophin expression and function, contributing to the development of neurocognitive disorders and neuropathogenesis.

In the central nervous system (CNS), HIV infection has been associated with decreased levels of mature BDNF and increased levels of proBDNF, the precursor form of BDNF. This imbalance suggests a disruption in the normal processing and release of BDNF, which can have significant implications for neuronal survival and plasticity. Neurons exposed to the HIV protein gp120, which is known to contribute to neurotoxicity, exhibit lower concentrations of mature BDNF and higher levels of proBDNF. This dysregulation of BDNF expression may contribute to neuronal dysfunction and cognitive impairments observed in HIV-infected individuals.

In the peripheral nervous system, HIV infection has also been linked to alterations in neurotrophin expression. Studies have shown that neurotrophins, including BDNF, can be upregulated in peripheral tissues during inflammatory conditions associated with HIV infection. This upregulation of BDNF expression in peripheral tissues, such as skin, muscle, or gut, may be a response to the inflammatory environment and serve as a protective mechanism. However, excessive or prolonged neurotrophin expression in peripheral tissues can contribute to inflammation and the upregulation of chemokines, which are involved in immune cell recruitment and activation.

Furthermore, HIV-associated neurotoxicity has been linked to the dysregulation of neurotrophins. The HIV proteins, such as gp120 and Tat, produced by HIV reservoirs in the brain, can modulate the expression of neurotrophins. For example, Tat has been shown to increase the expression of BDNF, as well as other proteins involved in inflammation and neuronal damage [11,12]. This dysregulated neurotrophin expression can lead to chronic inflammation and neuronal dysfunction, contributing to the pathogenesis of HIV-associated neurocognitive disorders.

Since up-to-date literature shows that neurotrophin expression, in particular, BDNF, was upregulated in invasive cervical cancer and that neurotrophins are also upregulated in the peripheral nervous system in patients with HIV infection, we aimed to investigate whether neurotrophins, in HIV-infected women, might play a crucial role in the enhancement and accelerated progression of cervical preneoplastic disease and invasive tumors. Therefore, this study aimed to evaluate the expression of neurotrophins NGF and BDNF in specimens from preneoplastic cervical lesions according to the HIV status of the patients.

## 2. Results

We included in our analysis 35 women matching the inclusion criteria: 12 HIV-infected patients who were matched with 23 HIV-negative patients. HIV patients and the control group did not differ in terms of mean age and parity, HPV-HR status, or indications for LEEP. Ten of twelve (83.3%) HIV-positive women were on ART at the time of inclusion in the study, whereas no HIV-negative patient was on ART. The most common indication for LEEP was persistent L-SIL or ASCUS, ASC-H at cytology for both HIV-positive (58.3%) and HIV-negative (56.5%) women. The most common histopathological finding in LEEP specimens was CIN 3/squamous carcinoma in situ in both groups (83.3% in HIV-positive women and 100% in the control group; Table 1).

Immunohistochemistry analysis showed that BDNF expression was significantly higher in cervical preneoplastic lesions from HIV-positive women than in cervical lesions from HIV-negative women. In particular, BDNF was expressed in 8/12 HIV-positive patients and in 7/23 HIV-negative patients (66.7% vs. 30.4%, χ^2^ = 4.227; *p* = 0.040; Figure 1 and Figure 2).

NGF expression was not significantly higher in cervical preneoplastic lesions from HIV-positive women than in cervical lesions from HIV-negative women. In particular, NGF was expressed in 8/12 HIV-positive patients and in 12/23 HIV-negative patients (66.7% vs. 52.2% χ^2^ = 0.676; *p* = 0.411; Figure 3 and Figure 4).

We tested the role of the HIV status in the expression of neurotrophins in preneoplastic cervical disease. Therefore, we performed a logistic regression analysis, including the following variables: HIV status, presence of HPV-HR, parity, indications for LEEP (1. persistent L-SIL or ASCUS, ASC-H at cytology; 2. H-SIL at cytology; 3. CIN 2/3 at the histology of cervical biopsies during a colposcopy exam), and histological diagnosis at LEEP (1. CIN 2; 2. CIN 3/squamous carcinoma in situ). Logistic regression analysis showed that the HIV status is an independent predictor of BDNF expression in pre-invasive cervical disease when considered alone (crude OR 4.6, 95% CI 0.027–20.347; *p* = 0.046) and when analyzed with other co-factors (adjusted OR 6.786, 95% CI 1.084–42.476; *p* = 0.041; Table 2).

## 3. Discussion

Our study demonstrated that BDNF expression in the preneoplastic cervical disease is higher in HIV-infected women compared with non-infected controls. We showed that this correlation is independent of the clinical features of the patients and the presence of the HPV-HR genotype. On the other hand, NGF expression did not show any difference according to the HIV status.

In the central nervous system (CNS), neurotrophins and their receptors, particularly Trk receptors, play a crucial role in various neuronal functions, such as cell survival, differentiation, synapse formation, and axonal growth [13,14,15,16]. BDNF, in particular, is transported like a neurotransmitter in the neurons of the dorsal root ganglion (DRG) [17]. The anterograde transport of BDNF serves multiple purposes, including rapid postsynaptic functions, providing a source of BDNF to astrocytes in injured brain areas, acting as a trophic agent for postsynaptic targets, and facilitating signaling between nerve terminals of different neurons [18].

BDNF also has a fundamental role in the development of visceral innervation, and its production significantly increases in inflammatory diseases of adult viscera [19,20,21]. BDNF and its receptor TRKB have direct and indirect roles in angiogenesis, with both neurotrophins modulating VEGF expression in different cellular models [22,23,24,25]. In cervical cancer cell lines, BDNF/TRKB enhances cell proliferation by activating ERK and AKT signaling pathways. This suggests that TRKB can potentially increase VEGF expression in cervical cancer, considering the association between BDNF/TRKB activation and VEGF expression in other models [26,27,28].

The association between HIV infection and neurocognitive disorders has been linked to neurotrophins, particularly BDNF [29]. HIV infection leads to the overexpression of neurotrophins also in the peripheral nervous system, and BDNF is believed to play a significant role in the development of HIV-associated neurotoxicity. While BDNF has a neurotrophic role in the central nervous system, preventing apoptosis in neurons and microglial cells, it also contributes to inflammation in peripheral tissues by upregulating chemokines.

BDNF expression upregulated by gp120, a protein associated with HIV, has been implicated in the development of HIV-associated pain through the Wnt/β-catenin signaling pathway [11]. Additionally, Tat, produced by macrophage HIV reservoirs in the brain, has been shown to increase the expression of certain genes, including C5, APBA1, and BDNF, while decreasing CRLF2. This leads to the production of inflammatory cytokines/chemokines and viral proteins, promoting inflammation and neuronal damage in HIV neuropathogenesis [12,30].

The transmembrane protein receptor tyrosine kinase B (TrkB) is a specific receptor for BDNF and is part of the neurotrophic factor receptor Trk family. While TrkB is crucial for the development and maturation of the nervous system, evidence suggests its involvement in promoting tumor formation and metastasis in certain malignancies. Upregulation of the BDNF/TrkB pathway has been found to promote epithelial–mesenchymal transition, migration, and invasion in cervical cancer. BDNF/TRKB increases cell proliferation through the activation of ERK and AKT signaling pathways. Moreover, the activated PI3K/AKT pathway by TrkB/BDNF can inhibit caspase-3 activation, leading to increased resistance to apoptosis. BDNF and TrkB expression is significantly elevated in various cancer cells, including cervical cancer, highlighting their potential as targets for therapeutic intervention to improve patient survival and develop new treatments for cervical cancer metastasis.

This is the first study to evaluate the immunohistochemistry expression of BDNF and NGF in the preneoplastic cervical disease according to HIV infection, possibly highlighting the role of BDNF as a mediator of cervical dysplastic and neoplastic progression, enhanced by HIV infection. 

Previous studies demonstrated the presence of an immunosuppressive tumor immune microenvironment (TIME) in cervical cancer, which may explain the inability of the immune system in women with cervical cancer to react against the tumor cells [31]. Recently, Yu et al. evaluated the role of BDNF in the TIME of lung adenocarcinoma, and they found that the higher expression of BDNF mRNA was associated with advanced pathological stage, increased risk of metastasis, and poor overall survival in these patients, characterizing BDNF as an unfavorable prognostic indicator in lung adenocarcinoma. In particular, in this study, BDNF expression had a positive correlation with macrophages and CD8 + T cells and a negative correlation with B cells, a significant positive correlation with M2 macrophages, and an inverse correlation with M1 macrophages. Although the role of BDNF in the TIME of cervical cancer has not been investigated yet, this evidence may suggest the role of BDNF in modulating the immune response also in patients with cervical cancer. In this context, the immune system depletion associated with HIV infection may exert a role in enhancing the immunosuppressive TIME driven by BDNF [32]. HIV predominantly infects and replicates within CD4+ T cells, which play a central role in orchestrating the immune response. As the infection progresses, the virus destroys a significant number of these cells, resulting in a substantial decline in their population. This depletion leads to a compromised immune system as the ability to initiate an immune response against cancer cells is severely impaired. Since we demonstrated that BDNF is more expressed in preneoplastic cervical lesions of HIV-infected women compared with non-infected controls, our results may indicate that BDNF is the main responsible for the increased rate of tumor initiation and progression in women with HIV infection. However, our study focused only on preneoplastic cervical lesions; therefore, further research is needed on the evaluation of BDNF with the TIME in invasive cervical lesions of HIV-infected women, possibly also evaluating the association of the neurotrophin expression with the tumor stage and the overall patients’ survival.

In our proposed model, HIV infection causes the overexpression of BDNF in cervical tissues. Increased BDNF expression can be responsible for the accelerated progression of HPV-related dysplastic lesions, explaining the reason for the increased incidence of cervical cancer cases, the reduced interval between HPV infection and the onset of invasive cervical cancer, and the increased incidence of recurrences among HIV-positive women.

Furthermore, preliminary data have linked the expression of neurotrophins, in particular NGF, to the presence of perineural invasion (PNI) in certain tumors, including also one study investigating this relationship in patients with cervical cancer [33]. This study demonstrated that high NGF and TrkA expression, but not p75NRT expression, is associated with PNI in cervical cancer, further highlighting the possible role of neurotrophin modulation in the progression of cervical cancer. Currently, no data are available on the role of BDNF in the PNI of cervical cancer, especially in women with HIV infection.

The main limitation of the present study is the limited number of included cases and the missing information regarding the follow-up of included patients. On the other hand, the number of included cases is consistent with the incidence of preneoplastic cervical cancer and HIV infection in our country.

These results are interesting in light of the recent interest in therapeutic strategies for cancers involving TRK inhibitors in malignancies with TRK fusions. TRKB receptors, also known as TrkB or NTRK2, play a significant role in cancer development and progression. TRKB is a member of the TRK family of receptor tyrosine kinases, and its activation by neurotrophins, particularly brain-derived neurotrophic factor (BDNF), promotes cell survival, proliferation, and differentiation. Aberrant TRKB signaling has been implicated in various malignancies, including those with TRK fusions, which occur when the NTRK genes fuse with other partner genes, leading to constitutive TRKB activation.

In cancers with TRK fusions, such as certain types of pediatric cancers and rare adult cancers, the fusion proteins result in the expression of chimeric TRK proteins that continuously activate downstream signaling pathways, driving uncontrolled cell growth and survival [34]. These TRK fusions represent an oncogenic driver event and have emerged as actionable targets for cancer therapy.

The presence of NTRK fusions in 0.19% of cases of breast cancer, involving both secretory and non-secretory breast cancer subtypes, and in 0.68% of cases of breast metaplastic carcinoma has been demonstrated [35]. In a retrospective study on 287 cases of triple-negative breast cancer (TNBC), 11.5% of the cases showed a positivity for NTRK fusions at IHC, even if the subsequent FISH confirmed only 13 (4.5%) positive cases [36]. Furthermore, a more comprehensive meta-analysis on the frequency of NTRK fusion in solid tumors showed that highest NTRK gene fusion frequencies were reported in the following cancers: infantile/congenital fibrosarcoma (90.56%, 95% CI 67.42–100.00), secretory breast cancer (92.87%, 95% CI 72.62–100.00), and congenital mesoblastic nephroma (21.52%, 95% CI 13.06–32.20). Lower frequencies were reported in non-small cell lung cancer (0.17%, 95% CI 0.09–0.25), colorectal adenocarcinoma (0.26%, 95% CI 0.15–0.36), cutaneous melanoma (0.31%, 95% CI 0.07–0.55), and non-secretory breast carcinoma (0.60%, 95% CI 0.00–1.50) [37].

Therapeutic strategies for malignancies involving TRK fusions have focused on TRK inhibitors, which are designed to selectively block the kinase activity of the TRKB receptor. These inhibitors can disrupt the oncogenic signaling cascades triggered by TRK fusions, thereby inhibiting tumor growth and progression. Several TRK inhibitors, such as larotrectinib and entrectinib, have shown remarkable efficacy in clinical trials, demonstrating durable responses and tumor regression in patients with TRK fusion-positive cancers.

The use of TRK inhibitors represents a paradigm shift in precision oncology, as these therapies target the underlying genetic alteration driving tumor growth rather than the tumor type or tissue of origin. Importantly, TRK inhibitors have demonstrated significant clinical activity across a wide range of tumor types, including both solid tumors and hematological malignancies, harboring TRK fusions. Additionally, TRK inhibitors have shown efficacy in pediatric patients, highlighting their potential as a treatment option for pediatric cancers driven by TRK fusions.

To identify patients who may benefit from TRK inhibitors, molecular profiling and genetic testing methods are employed to detect TRK fusions. These tests assess the presence of NTRK gene fusions or TRKB protein expression and are used to guide treatment decisions. TRK fusions can be identified by DNA-based or RNA-based next-generation sequencing or other assays such as fluorescence in situ hybridization (FISH) or immunohistochemistry [38]. Larotrectinib is a small molecule that binds to NT receptors, thereby preventing neurotrophin–TRK interaction and TRK activation, which results in the induction of cellular apoptosis and the inhibition of cell growth. Larotrectinib has become the first tyrosine kinase inhibitor to be granted approval by the US Food and Drug Administration and the European Medicines Agency for a tumor-agnostic indication [39,40].

Given that angiogenesis is a key feature in gynecological neoplasms, and NTs act as direct and indirect angiogenic factors, it may be relevant to study whether TRK inhibitors can improve the efficacy of antiangiogenic drugs, especially in cervical cancers with a co-existing HIV infection. 

Presently, no data are available on the efficacy of NTRK inhibitors in HIV-infected patients with cancer, especially those diagnosed with cervical cancer.

## 4. Materials and Methods

Patients were prospectively recruited from the regional reference center for infectious diseases in obstetrics and gynecology at the Gynecology Department of the University of Naples “Federico II” from 2018 to 2019. Patients with the following inclusion criteria were enrolled: age between 15 and 50 years old; BMI < 30 kg/m^2^; no other comorbidities apart from HIV infection; and cervical intraepithelial neoplasia (CIN) grade 2 or 3 at the histological diagnosis of specimens from Loop Electrosurgical Excision Procedure (LEEP) for suspected cervical cancer (abnormal Pap smear, abnormal colposcopy findings, abnormal histological findings at cervical biopsy). We collected the following clinical data: age, parity, HIV infection stage, antiretroviral therapy (ART), indication for LEEP, and high-risk HPV (HPV-HR) genotyping. All included women underwent a transvaginal ultrasound for the assessment of the uterus, ovaries, and inguinal lymph nodes: only women with normal ultrasound findings were included in the study.

Indications for LEEP were grouped as follows: 1. persistent L-SIL or ASCUS, ASC-H at cytology; 2. H-SIL at cytology; and 3. CIN 2/3 at the histology of cervical biopsies during a colposcopy exam. LEEP histological results were grouped as follows: CIN 2 or CIN 3/squamous carcinoma in situ.

LEEP samples were sent for histological analysis already fixed in 10% formaldehyde and oriented by the gynecologist. Firstly, length, width, and wall thickness were measured for each LEEP sample; any grossly visible lesion was recorded in terms of length, width, and thickness; lesion location and its distance from margins were documented. Then, the deep margin was inked, and the full sample was submitted for histologic examination. Specimens were radially thinly sectioned (ideally 2 mm sections), ensuring each section had endocervical and ectocervical margins. They were serially submitted in a clockwise direction by putting one or two sections per cassette, depending on the size. Representative slides of the lesions were stained with hematoxylin and eosin. LEEP sections then were evaluated under the microscope by a pathologist for diagnosis and choice of the most appropriate sections for further protein expression evaluation (Figure 5).

### 4.1. Neurotrophin Expression in Cervical Dysplastic and Neoplastic Lesions

NGF and BDNF expression was evaluated using immunohistochemistry (IHC). Initially, one or two sections per patient were chosen among the others by considering morphological appearances at H&E staining: only sections with dysplastic lesions of cervical epithelium were included, and, for each patient, slides containing larger stromal portions were considered more suitable for the experiment in order to have an internal positive control. Human brain tissue slides were used as control of the entire procedure and for specific protocol suiting. Immunohistochemistry was executed with Benchmark Ultra Roche Ventana automated system. Samples were deparaffinated at 72 °C. Heat-induced epitope retrieval was performed at 100 °C for 54 min in a buffer solution with basic pH (7.8). Anti-BDNF and anti-NGF antibodies (Invitrogen, Rabbit Polyclonal Antibody) were incubated for 48 min at 37 °C with a dilution of 1:100. For signal detection procedure, an OptiView DAB IHC detection kit was used. In the end, counterstain was performed with hematoxylin for 8 min. Antibody detection was performed using a multilink streptavidin–biotin complex method, and antibodies were visualized by a diaminobenzidine chromagen method. Negative control samples were incubated with primary antibodies only. 

NGF and BDNF showed cytoplasmatic dot-like expression. In the analysis, this characteristic was taken into consideration, and the evaluating score was created considering both the signal intensity and the number of cells from the lesion that expressed the neurotrophin. Only cells with definite cytoplasmic staining were considered positive for each antibody. In particular, a score of 0–3 was given: 0, negative staining; 1, weak expression of the cytoplasm of tumor cells; 2, moderate expression of tumor cells; and 3, strong expression of tumor cells. A score of 0 or 1 was considered negative, and a score of 2 or 3 was considered positive.

### 4.2. Statistical Analysis

Patients were grouped according to the HIV status expression, and clinical parameters were compared between the two groups. The Shapiro–Wilk test was performed to test for normality. When normally distributed, continuous variables were compared by Student’s *t*-test, otherwise by the Kruskal–Wallis test; categorical variables were compared by the Pearson chi-square test. Furthermore, differences in the expression of NGF and BDNF were evaluated between the two groups. A *p*-value < 0.05 was considered significant. Statistical analysis was carried out using the Statistical Package for Social Sciences (SPSS) Statistics v. 19 (IBM Inc., Armonk, NY, USA).

## 5. Conclusions

BDNF is overexpressed in the preneoplastic cervical disease in HIV-positive women compared with HIV-negative controls. Our results show that BDNF can play a key role as a link between the pathways by which HIV and HPV interact to accelerate cervical cancer progression and invasion. These data can be useful to better understand the role of neurotrophins in the pathology of cervical cancer and the possible therapeutic strategies to improve disease outcomes.

## Figures and Tables

**Figure 1 ijms-24-10729-f001:**
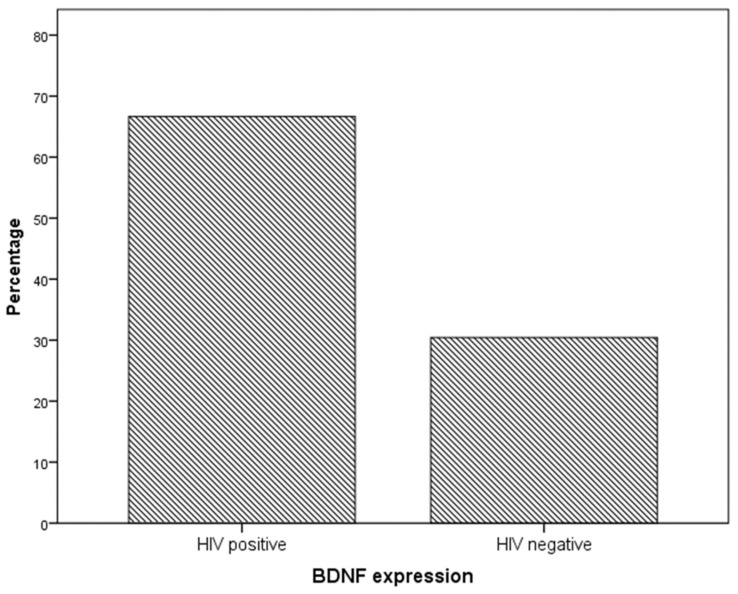
BDNF expression in specimens from the cervical preneoplastic disease according to the HIV status.

**Figure 2 ijms-24-10729-f002:**
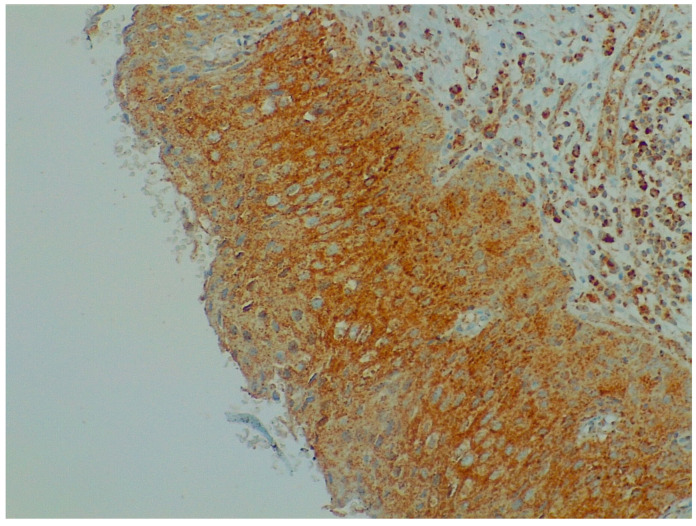
Immunohistochemistry: BDNF expression in cervical preneoplastic disease.

**Figure 3 ijms-24-10729-f003:**
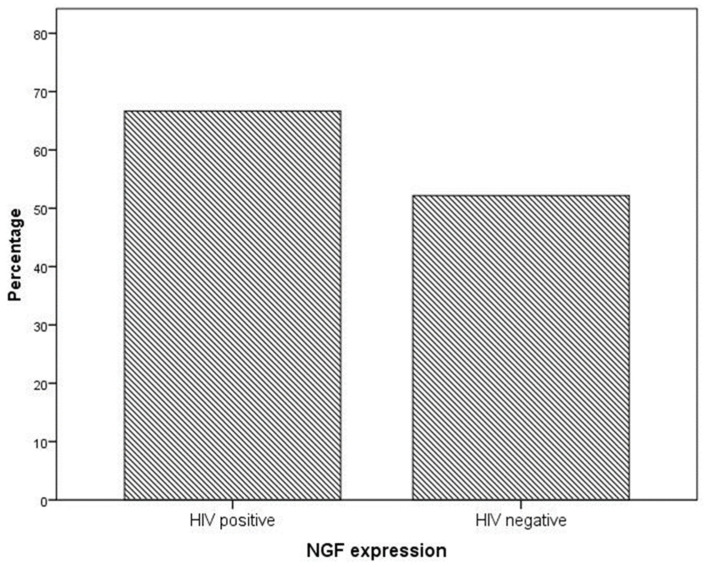
NGF expression in specimens from the cervical preneoplastic disease according to the HIV status.

**Figure 4 ijms-24-10729-f004:**
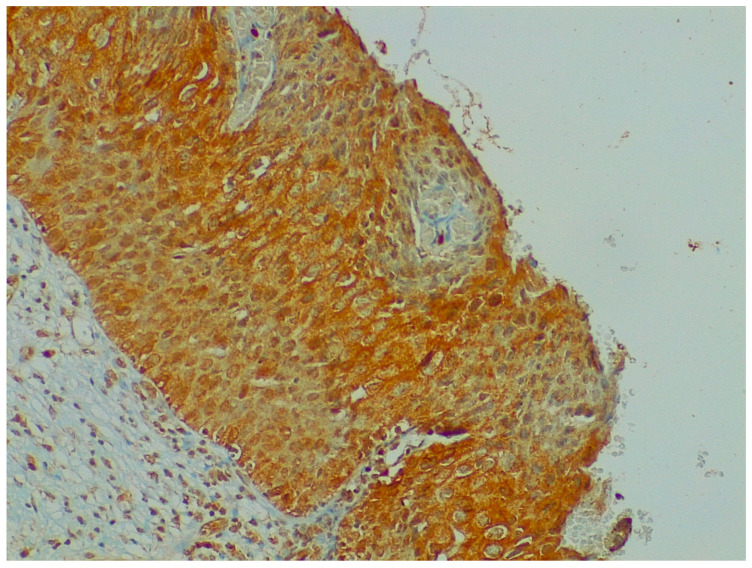
Immunohistochemistry: NGF expression in cervical preneoplastic disease.

**Figure 5 ijms-24-10729-f005:**
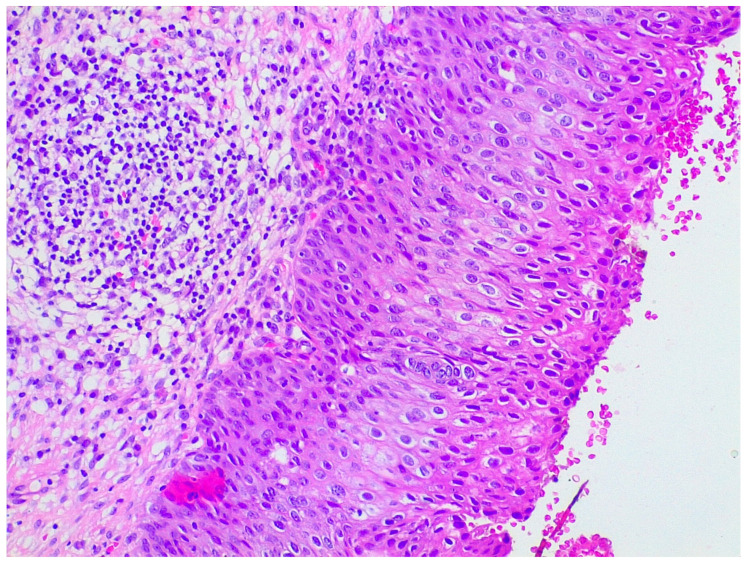
Hematoxylin and eosin staining of a patient with preneoplastic cervical disease.

**Table 1 ijms-24-10729-t001:** Characteristics of included patients.

Characteristics	HIV + (n = 12)	HIV − (n = 23)	*p*
Age (years)	41.83 (3.23)	40.17 (2.28)	0.676
Parity	1.25 (0.96)	0.69 (0.97)	0.118
HPV-HR positive	8 (66.7%)	11 (47.8%)	χ^2^ = 1.128; *p* = 0.288
Indication for LEEP *	1. 7 (58.3%) 2. 2 (16.7%) 3. 3 (25%)	1. 13 (56.5%) 2. 6 (26.1%) 3. 4 (17.4%)	χ^2^ = 0.539; *p* = 0.764
LEEP histological evaluation °	1. 2 (16.7%) 2. 10 (83.3%)	1. 0 (0%) 2. 23 (100%)	χ^2^ = 4.066; *p* = 0.044

Data are given as mean; standard deviation or n (%). HIV: human immunodeficiency virus; HPV-HR: human papillomavirus high-risk genotype; LEEP: Loop Electrosurgical Excision Procedure. * 1. Persistent L-SIL or ASCUS, ASC-H at cytology; 2. H-SIL at cytology; 3. CIN 2/3 at the histology of cervical biopsies during a colposcopy exam. ° 1. CIN 2; 2. CIN3/squamous carcinoma in situ.

**Table 2 ijms-24-10729-t002:** Logistic regression analysis: unadjusted and adjusted odds ratios of variables associated with BDNF expression.

	OR	95% CI	*p*	aOR	95% CI	*p*
Parity	1.231	0.621–2.442	0.551	1.063	0.486–2.323	0.879
HIV	4.571	1.027–20.347	0.046	6.786	1.084–42.476	0.041
HPV-HR positive	0.935	0.244–3.584	0.922	0.532	0.104–2.728	0.449
Indication for LEEP	0.765	0.323–1.809	0.541	0.563	0.198–1.600	0.281
LEEP histological evaluation	0.737	0.042–12.821	0.834	3.001	0.115–77.999	0.509

HIV: human immunodeficiency virus; HPV-HR: human papillomavirus high-risk genotype; LEEP: Loop Electrosurgical Excision; OR: odds ratio; aOR: adjusted odds ratio; 95th CI: 95th confidence interval.

## Data Availability

Data are available upon request to the corresponding author.

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
