# Peer review of "BDNF and NGF Expression in Preneoplastic Cervical Disease According to HIV Status"

_ijms, 2023, doi:10.3390/ijms241310729_

Round 1

Reviewer 1 Report

The topic of the manuscript is intriguing given the emerging importance of TRkA/B and C in the field of cancer.

However, there are a lot of concerns:

1) the data are not well contextualized. Despite it is clear that there is a correlation between cervical cancer and neurotrophin and their cognate receptors, the link between the virus and neurotrophins is not well explained and introduced.

This leaves open the questions: why do the authors have analyzed this aspect?

If they analyze this aspect gor correlating cervical cancer with neurotrophins expression, it remains a topic already investigated.

They should clarify.

2) Also, if the pictures are well-performed, they are only preliminary results. 

There is not a biological correlation and/or a molecular insight.

In this form, the paper is not suitable for the publication.

There are only some typos

Author Response

We thank the Editor and reviewers for their suggestions that helped us to improve our manuscript. Here we provide point-by-point answer to the reviewers. We increased the manuscript word count and we also checked our manuscript for duplication in order to reduce duplicated sentences, even if some sentences where we cite previous studies are referenced to the original papers. Corrections will be highlighted in yellow in the manuscript.

Reviewer 1

The topic of the manuscript is intriguing given the emerging importance of TRkA/B and C in the field of cancer.

However, there are a lot of concerns:

1) the data are not well contextualized. Despite it is clear that there is a correlation between cervical cancer and neurotrophin and their cognate receptors, the link between the virus and neurotrophins is not well explained and introduced.

This leaves open the questions: why do the authors have analyzed this aspect?

If they analyze this aspect gor correlating cervical cancer with neurotrophins expression, it remains a topic already investigated.

They should clarify.

We thank the reviewer for this consideration. HIV-infected women as a special population, are at increased risk for cervical cancer. Although the link between HIV infection and cervical cancer is confirmed from published epidemiological studies, the exact biological mechanisms underlying the role of HIV infection in the initiation and progression of cervical preneoplastic disease are not yet established (Lines 58-61). Therefore, we aimed to investigate the possible biological mechanisms that drive the association between HIV and cervical preneoplastic disease. Recent preliminary published studies have demonstrated that HIV infection is associated with the upregulation of neurotrophins mediated by HIV proteins. Therefore, in light of the recent evidence about the role of neurotrophins in the etiopathogenesis of cervical cancer, we hypothesized a possible direct effect of HIV infection in the expression of neurotrophins in cervical preneoplastic disease. (Lines 91-118).

2) Also, if the pictures are well-performed, they are only preliminary results. 

There is not a biological correlation and/or a molecular insight.

In this form, the paper is not suitable for the publication.

we agree with reviewers that pictures chosen in the first version did not correlate well with the results of our study; therefore, after an accurate selection, we provided new pictures of IHC analysis for both BDNF and NGF (Figures 2 and 4). Futhermore, we also added an Hematoxylin-Eosin section from an included case. We hope that these changes made the manuscript more interesting for the readers (Figure 5).

Reviewer 2 Report

The paper entitled «BDNF and NGF expression in preneoplastic cervical disease according to HIV status  » by Angelo Sirico et al., reports that BDNF expression level, assessed by immunohistochemistry, was more frequently observed in CIN2/3 lesions developed in HIV positive women  (8/12) than in HIV negative (7/23 ) patients ( p=0.040). The authors make the hypothesis that BDNF may act in the progression of CIN in HIV positive patients.

Main remarks

1)      The main result looks to be fragile. No details are provided on the methodology of analysis that is supposed to combine the number of positive cells and their staining intensity. The figure 2 is not convincing for a high-grade cervical lesion. Since there was 8 positive cases, I guess it is possible to find an unequivocal case to illustrate this positivity. By contrast, the NGF expression (no significant difference between the two groups) looks to be strongly positive (Fig 4), but only in the gland of the stroma whereas the malpighian epithelium (high grade?) is negative. Is it a case of glandular neoplasia ? Please explain.

2)      In HIV positive women, the immune response is hampered and this is supposed to account for the frequent and rapid progression of CIN in these patients. What could be the link between BDNF expression and the quality of the immune response ?

            In conclusion the paper should be improved by a better choice of the figures that illustrate the main data of the work. Adequate comments should help the reader to understand the meaning of these histological pictures. The discussion should take into account the deficience of the immune system in HIV patients and guide the reader to the possible role of BDNF in this context.

Correct English

Author Response

We thank the Editor and reviewers for their suggestions that helped us to improve our manuscript. Here we provide point-by-point answer to the reviewers. We increased the manuscript word count and we also checked our manuscript for duplication in order to reduce duplicated sentences, even if some sentences where we cite previous studies are referenced to the original papers. Corrections will be highlighted in yellow in the manuscript.

Reviewer 2

The paper entitled «BDNF and NGF expression in preneoplastic cervical disease according to HIV status  » by Angelo Sirico et al., reports that BDNF expression level, assessed by immunohistochemistry, was more frequently observed in CIN2/3 lesions developed in HIV positive women  (8/12) than in HIV negative (7/23 ) patients ( p=0.040). The authors make the hypothesis that BDNF may act in the progression of CIN in HIV positive patients.

Main remarks

1)      The main result looks to be fragile. No details are provided on the methodology of analysis that is supposed to combine the number of positive cells and their staining intensity. The figure 2 is not convincing for a high-grade cervical lesion. Since there was 8 positive cases, I guess it is possible to find an unequivocal case to illustrate this positivity. By contrast, the NGF expression (no significant difference between the two groups) looks to be strongly positive (Fig 4), but only in the gland of the stroma whereas the malpighian epithelium (high grade?) is negative. Is it a case of glandular neoplasia ? Please explain.

we agree with reviewers that pictures chosen in the first version did not correlate well with the results of our study; therefore, after an accurate selection, we provided new pictures of IHC analysis for both BDNF and NGF (Figures 2 and 4). Futhermore, we explained in details the procedure of histological and IHC evaluation (323-390). we also added an Hematoxylin-Eosin section from an included case. We hope that these changes made the manuscript more interesting for the readers (Figure 5).

2)      In HIV positive women, the immune response is hampered and this is supposed to account for the frequent and rapid progression of CIN in these patients. What could be the link between BDNF expression and the quality of the immune response ?

            In conclusion the paper should be improved by a better choice of the figures that illustrate the main data of the work. Adequate comments should help the reader to understand the meaning of these histological pictures. The discussion should take into account the deficience of the immune system in HIV patients and guide the reader to the possible role of BDNF in this context.

We agree with the reviewer and we added a paragraph speculating the possible link between BDNF expression and the quality of the immune response in HIV patients in light of the etiopathogenesis of cervical cancer at  Lines 227-252.

Round 2

Reviewer 1 Report

At this point, the authors should add:

1) details about the role of the Trks family in other types of cancer (e.g prostate, breast (and/orTNBC);

2) Since NGF is able to promote perineural invasion (PNI), is there evidence about the PNI for cervical cancer?

ok

Author Response

Reviewer 1

At this point, the authors should add:

1) details about the role of the Trks family in other types of cancer (e.g prostate, breast (and/orTNBC);

We thak the reviewer for the suggestion. We added details about the role of TRKS family in other tumors at lines 282-294

2) Since NGF is able to promote perineural invasion (PNI), is there evidence about the PNI for cervical cancer?

We thank the reviewer for the suggestion. Currently there is only limited evidence for the role of neurotrophins, in particular NGF, in the PNI of cervical cancer, we briefly discussed this point at Lines 258-264

Reviewer 2 Report

Histological pictures  OK

Paragraphs 4.1 and 4.2 should be discarded since it correspond to the routine technics of Pathology

No specific remark

Author Response

Reviewer 2

Histological pictures  OK

Paragraphs 4.1 and 4.2 should be discarded since it correspond to the routine technics of Pathology

We thank the reviewer. We deleted paragraphs 4.1 and 4.2 accordingly.